# Multi-omics investigation of *Clostridioides difficile*-colonized patients reveals pathogen and commensal correlates of *C. difficile* pathogenesis

Skye RS Fishbein[1,2], John I Robinson[3], Tiffany Hink[4], Kimberly A Reske[4], Erin P Newcomer[1,2], Carey-Ann D Burnham[2,5,6], Jeffrey P Henderson[3], Erik R Dubberke[4]*, Gautam Dantas[1,2,5,7]*

[1]The Edison Family Center for Genome Sciences and Systems Biology, Washington University School of Medicine, St Louis, United States; [2]Department of Pathology and Immunology, Washington University School of Medicine, St. Louis, United States; [3]Center for Women's Infectious Disease Research, Division of Infectious Diseases, Department of Internal Medicine, Washington University School of Medicine, St Louis, United States; [4]Division of Infectious Diseases, Washington University School of Medicine, St. Louis, United States; [5]Department of Molecular Microbiology, Washington University School of Medicine, St Louis, United States; [6]Department of Pediatrics, Washington University School of Medicine, St. Louis, United States; [7]Department of Biomedical Engineering, Washington University in St. Louis, St. Louis, United States

*For correspondence:
edubberk@wustl.edu (ERD);
dantas@wustl.edu (GD)

**Abstract** *Clostridioides difficile* infection (CDI) imposes a substantial burden on the health care system in the United States. Understanding the biological basis for the spectrum of *C. difficile*-related disease manifestations is imperative to improving treatment and prevention of CDI. Here, we investigate the correlates of asymptomatic *C. difficile* colonization using a multi-omics approach. We compared the fecal microbiome and metabolome profiles of patients with CDI versus asymptomatically colonized patients, integrating clinical and pathogen factors into our analysis. We found that CDI patients were more likely to be colonized by strains with the binary toxin (CDT) locus or strains of ribotype 027, which are often hypervirulent. We find that microbiomes of asymptomatically colonized patients are significantly enriched for species in the class Clostridia relative to those of symptomatic patients. Relative to CDI microbiomes, asymptomatically colonized patient microbiomes were enriched with sucrose degradation pathways encoded by commensal Clostridia, in addition to glycoside hydrolases putatively involved in starch and sucrose degradation. Fecal metabolomics corroborates the carbohydrate degradation signature: we identify carbohydrate compounds enriched in asymptomatically colonized patients relative to CDI patients. Further, we reveal that across *C. difficile* isolates, the carbohydrates sucrose, rhamnose, and lactulose do not serve as robust growth substrates in vitro, consistent with their enriched detection in our metagenomic and metabolite profiling of asymptomatically colonized individuals. We conclude that pathogen genetic variation may be strongly related to disease outcome. More interestingly, we hypothesize that in asymptomatically colonized individuals, carbohydrate metabolism by other commensal Clostridia may prevent CDI by inhibiting *C. difficile* proliferation. These insights into *C. difficile* colonization and putative commensal competition suggest novel avenues to develop probiotic or prebiotic therapeutics against CDI.

## Editor's evaluation

Not everyone colonized by *C. difficile* has gut symptoms, but the reasons why are unclear. This article uses the combination of sequencing and mass spectrometry to compare patients with or without symptoms, revealing links between specific gut bacteria and diet, which could lead to diet or bacterial treatment or prevention strategies.

## Introduction

*Clostridioides difficile* infection (CDI) remains a significant cause of morbidity and mortality in the health care setting and in the community (*Guh et al., 2020*). Antibiotic treatments, among other risk factors associated with weakened colonization resistance, increase susceptibility to CDI (*Dubberke and Olsen, 2012*; *Eze et al., 2017*). *C. difficile* residence in the human gastrointestinal (GI) tract may result in a spectrum of clinical manifestations, from asymptomatic colonization to severe CDI-related colitis and fatal toxic megacolon (*Crobach et al., 2018*). Diagnosis of CDI relies on detection of the protein toxin, most commonly by enzyme immunoassay (EIA), or the detection of the toxin-encoding genes *tcdA* and *tcdB*, by nucleic acid amplification test (NAAT). These diagnostic tools serve as rough benchmarks for assessing the severity of disease. Discrepancies between the results of these assays, as in the case of patients with clinically significant diarrhea (CSD) who are EIA negative (EIA-) but NAAT positive for toxigenic *C. difficile* (Cx+), highlight the complexity of states in which *C. difficile* can exist in the GI tract. Because CDI is a multi-factorial interaction between the host, pathogen, and microbiome, clarifying the differences in biological correlates between asymptomatic colonization (Cx+/EIA-) and CDI (Cx+/EIA+) is critical for identifying mechanisms of colonization resistance, and for defining novel probiotic or prebiotic avenues for treatment or prevention of CDI (*Kondepudi et al., 2012*; *Rätsep et al., 2017*).

*C. difficile* enters the GI tract as a spore, germinates in the presence of primary bile acids, and replicates through consumption of amino acids and other microbiota or host-derived nutrients (*Hryckowian et al., 2017*). Notably, many of these metabolic cues are characteristic of a perturbed microbiome (*Nagao-Kitamoto et al., 2020*; *Battaglioli et al., 2018*). The hallmark of *C. difficile* pathogenesis is the expression of the toxin locus encoded on the *tcd* operon; this locus is tightly regulated by nutrient levels (*Martin-Verstraete et al., 2016*). Correspondingly, it is hypothesized that an environment replete of nutrients induces toxinogenesis, allowing *C. difficile* to restructure the gut environment and acquire nutrients through inflammation (*Fletcher et al., 2018*; *Fletcher et al., 2021*). The instances of patients who are colonized but have no detectable *C. difficile* toxin in their stool suggests that these patients' microbiomes may be less permissive towards CDI development. Identification of metabolic traits within the microbiome of asymptomatic, *C. difficile*-colonized patients could reveal a number of potential therapeutic pathways toward precise amelioration of symptomatic *C. difficile* disease.

A multitude of probiotic and prebiotic approaches have demonstrated efficacy to curb *C. difficile* proliferation in vivo (*Rätsep et al., 2017*; *Chen et al., 2020*; *Pereira et al., 2020*). While restoration of the microbiota through fecal microbiota transplantation can provide colonization resistance (*Laffin et al., 2017*), the molecular mechanisms of how this resistance is conferred remain unclear. Recent studies using a murine model of infection have indicated that the administration of carbohydrates (both complex and simple) in the diet can be used to curb or prevent CDI (*Mefferd et al., 2020*; *Schnizlein et al., 2020*; *Hryckowian et al., 2018*). Paradoxically, integrated metabolomics and transcriptomics data collected during murine *C. difficile* colonization indicates that simple carbohydrates are imperative for pathogen replication (*Fletcher et al., 2018*). It is critical to understand the mechanism by which catabolism of specific carbohydrates could inhibit *C. difficile* proliferation in the human GI tract.

Here, we perform a multi-level investigation of two relevant patient populations, those colonized with *C. difficile* but EIA negative (asymptomatically colonized) and those who are EIA positive (CDI) to understand the microbial and metabolic features that may underlie protection from CDI. First, we use microbiome analyses to identify a number of non-*C. difficile*, clostridial species that are negatively correlated with *C. difficile* in asymptomatically colonized individuals. Secondly, interrogation of a metabolomics dataset from the same patient population (*Robinson et al., 2019*) reveals increased abundance of a number of carbohydrate metabolites in asymptomatically colonized patients. Finally, we show that some metabolites enriched in asymptomatically colonized individuals are largely

non-utilizable by *C. difficile* isolates. Together, these datasets reveal that asymptomatically colonized patients are defined by an interaction of clostridial species and carbohydrate metabolites that may serve as a last-line of resistance against CDI in colonized patients.

## Results

The clinical manifestation of *C. difficile* colonization in a host gastrointestinal tract is determined by a multi-factorial interaction between the host, their microbiome, and the pathogen. We hypothesized that, among these factors, natural variation in *C. difficile* strains infecting patients might differentiate asymptomatic from CDI patients (*Dubberke et al., 2018*). Through retrospective analysis of a human cohort of 124 patients (*Supplementary file 1*) with clinically significant diarrhea (CSD) and stool submitted for *C. difficile* toxin testing, we defined two cohorts: those diagnosed with CDI (Cx+/EIA+) or those asymptomatically colonized (Cx+/EIA-) (*Robinson et al., 2019*). EIA status (EIA+ or EIA-) was determined by the result of the clinical toxin EIA performed on the stool specimen, and a positive toxigenic culture (a *C. difficile* isolate with with *tcdA* and/or *tcdB*; Cx+) (*Dubberke et al., 2018*). In-depth analysis of *C. difficile* isolate factors related to EIA status was performed on the isolates corresponding to the 102 metagenomic samples analyzed (see Materials and Methods, *Supplementary file 2*). Multiplex PCR was used to identify isolates with *cdtAB*, the binary toxin locus (*Cowardin et al., 2016*). Notably, there was a significant enrichment of isolates with *cdtAB* in the stools of patients with CDI (*Figure 1A*; p = 0.0012, Fisher's exact test). Additionally, there were differences in the distribution of *C. difficile* strains associated with the two patient cohorts; CDI patients were more likely to be infected by a *C. difficile* isolate of the ribotype 027 lineage (*Figure 1A*; p = 0.0058, Fisher's exact test), a clade likely to contain virulent members (*Merrigan et al., 2010*). Interestingly, of the isolates positive for *cdtAB* (22 out of 102 isolates), 36% were considered a ribotype 027 strain. Given these genetic indicators of potential differences in virulence, we asked if strains from both groups were capable of producing toxin, using culture supernatants from in vitro broth culture. We found that 56% of isolates expressed detectable TcdA/B, with no significant different (p = 0.86) in the capacity of strains from Cx+/EIA- stools (24 out of 54 isolates) or Cx+/EIA+ (24 out of 48 isolates) to elaborate toxin (*Figure 1—figure supplement 1A*). Predictably, differences in genetic indicators of strain virulence (as indicated by prevalence of both a prominent ribotype and a second toxin locus) were significant correlates of EIA status.

As antibiotics are a well-known risk factor for CDI, we analyzed previous inpatient antibiotic orders (within one month prior to diagnosis) for patients in the Cx+/EIA- and Cx+/EIA+ cohort, as a proxy for antibiotic exposure (*Table 1*). Fitting antibiotic exposure to a logistic regression model (McFadden's $R_2$ = 0.306) revealed that CDI was significantly associated with cephalosporin exposure. Analysis of potential antibiotic exposures in our patient cohort confirms the risk that antibiotics pose for CDI development (*Mullish and Williams, 2018*; *Webb et al., 2020*).

Antibiotics increase susceptibility to CDI through disruption of colonization resistance, mainly conferred to the host via the gut microbiome (*Theriot et al., 2014*). To determine the microbial correlates of disease state, we performed shotgun metagenomic sequencing on patient stool samples from the asymptomatic (n = 54) and CDI (n = 48) groups, and classified species using MetaPhlAn2. Given the strong association with antibiotic exposure in our CDI cohort, we hypothesized that our asymptomatically colonized patients would have increased microbiome-mediated colonization resistance relative to CDI patients. We examined community structure in stool metagenomes and found that there was no significant difference in Faith's diversity (*Faith and Baker, 2007*), a measure of alpha-diversity that incorporates phylogenetic relationships, between patient groups (*Figure 1—figure supplement 1B*, Wilcoxon rank-sum test, p = 0.1602). There were no significant differences in beta-diversity, as measured by weighted Unifrac distance, between EIA status (p = 0.233, permutational analysis of variance test [PERMANOVA]) (*Figure 1B*). Although we hypothesized that increased virulence (through additional toxin allele or ribotype) associated with EIA+ could affect microbiome structure, we found no significant association between beta-diversity and *cdtAB* presence (*Figure 1—figure supplement 1C*; p = 0.799, PERMANOVA) or ribotype distribution (*Figure 1—figure supplement 1D*; p = 0.982, PERMANOVA). Previous comparative microbiome studies have revealed phylum-level differences in Bacteroides and Firmicutes in CDI cases versus controls not colonized with *C. difficile* (*Kachrimanidou and Tsintarakis, 2020*). In contrast, we found no significant differences in relative abundance of bacterial phyla between asymptomatically colonized patients and patients with

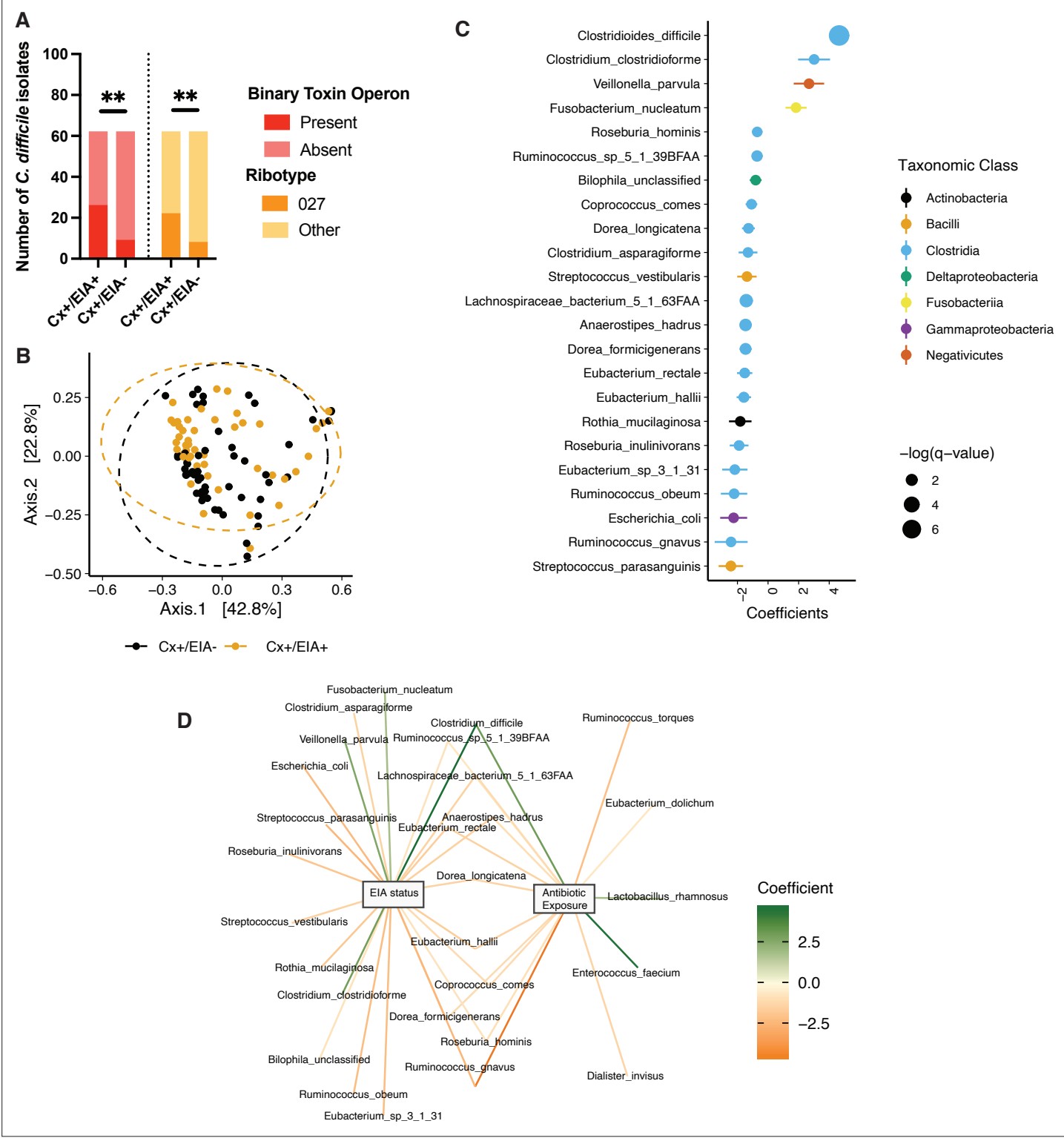

**Figure 1.** Pathogen and microbiome determinants of *C. difficile*-colonized patients. (**A**) *Clostridioides difficile* isolate distribution based on PCR and ribotyping data for each isolate cultured from patient stools. **, p < 0.001 as measured by a Fisher's exact test. (**B**) Principal coordinate analysis (PCoA) of weighted Unifrac distances between stool microbiomes. Colors indicate EIA status. Groups were not significantly different as measured by a PERMANOVA (p = 0.69). (**C**) Significant microbial taxa associated with disease state, where a positive coefficient is associated with Cx+/EIA+ state and a negative coefficient is associated with Cx+/EIA- state. Colors indicate taxonomic Class of the microbial feature, and the size of circle corresponds to magnitude of statistical significance. Features with q-value of <0.25 were plotted. (**D**) Network of features associated with antibiotic exposure or EIA

*Figure 1 continued on next page*

*Figure 1 continued*

status. Species nodes are connected to metadata nodes by edged colored with the feature weight (coefficient) computed using linear mixed modeling (MaAslin2). All taxa displayed had a q-value of <0.25 in respective analyses.

The online version of this article includes the following source data and figure supplement(s) for figure 1:

**Source data 1.** Raw absorbance value for in vitro toxin ELISA of 102 *C.difficile* isolates.

**Figure supplement 1.** Microbiome configuration of *C. difficile*-colonized patients.

**Figure supplement 2.** Kraken analysis of metagenomic data.

CDI (*Figure 1—figure supplement 1E*). These data indicate that there were no gross differences in microbiome structure related to either EIA status or pathogen features.

Instead, we hypothesized that differences between these states may manifest at higher resolution. We used a multivariable regression model, as implemented by MaAslin2 (*Mallick et al., 2021*) to identify microbial taxa predictive of either group. Interestingly, species from class Clostridia were most enriched in taxa significantly altered by EIA status (Fisher's exact test, p = 0.0022). *C. difficile* was the strongest predictor of CDI state, whereas non-*C. difficile* clostridial taxa were predictive of asymptomatic state (*Figure 1C*, FDR < 0.25). Correspondingly, we saw increased *C. difficile* relative abundance in CDI patients and increased levels of a number of non-*C. difficile* clostridial species, including *Eubacterium* spp., *Dorea* spp., and *Lachnospiraceae* spp. in asymptomatic patients (*Figure 1—figure supplement 1F*). Given our inability to detect *C. difficile* in all sequenced stools (70 out of 102 culture-positive stool samples), we utilized an alternative metagenomic species classifier Kraken (*Wood and Salzberg, 2014*), to validate our findings. Using Kraken, we detected *C. difficile* in nearly all stool metagenomes (101 out of 102). Using the identical linear mixed modeling approach (MaAsLin2), we recapitulated data indicating that *C. difficile* abundance was the strongest predictor of EIA status and increased in Cx+/EIA+ patients. Additionally, a number of commensal clostridial taxa from the *Eubacterium* genus and *Anaerostipes* genus were strongly associated with EIA- status, confirming prior MetaPhlAn2 predictions (*Figure 1—figure supplement 2A*,B).

Using our microbiome data, we examined the association between *C. difficile* levels and pathogen markers previously associated with EIA status. We found that *C. difficile* relative abundance was not significantly different when stratified by isolate CDT status (*Figure 1—figure supplement 2C*; p = 0.3, Wilcoxon rank sum) or isolate ribotype (p = 0.78, Kruskal-Wallis). Notably, there was a slight, yet insignificant increase in *C. difficile* abundance in microbiomes associated with a ribotype 027 isolate (*Figure 1—figure supplement 2D*) relative to microbiomes associated with *C. difficile* isolates of other ribotypes. We also interrogated taxonomic features that were predictive of antibiotic exposure. Expectedly, we found that taxonomic features predictive of CDI state were also associated with antibiotic exposure (*Figure 1D*). Our data indicate that patients with asymptomatic *C. difficile* colonization or CDI do not have grossly different gut microbiome community structures but instead have distinctive alterations in a subset of species from class Clostridia and class Bacilli in the microbiota.

*C. difficile* pathogenesis is heavily affected by carbohydrate, amino acid, and bile acid levels in the gastrointestinal tract, related to the metabolism of competitive commensals (*Sorbara and Pamer, 2019*). To identify metabolic pathways in other clostridia that might enable them to outcompete *C. difficile*, we defined metabolic potential in patient microbiomes using HUMAnN2 to quantify microbial pathway abundances. We found no significant differences in alpha- or beta-diversity between overall metabolic pathway composition in the two patient microbiome groups (*Figure 2—figure supplement 1A*,B; p = 0.2393, Wilcoxon rank sum and p = 0.054, PERMANOVA). Therefore, we trained an elastic net model to identify specific pathways associated with EIA status (*Figure 2A*). We found a number of carbohydrate degradation pathways and amino acid biosynthetic pathways

**Table 1.** Logistic regression coefficients for antibiotic exposures associated with Cx+/EIA+ in patient cohort.

| Antibiotic | Coefficient | Standard error | p-Value |
|---|---|---|---|
| Cephalosporin | 2.68 | 0.74 | 2.70E-04 |
| Fluoroquinolone | 0.19 | 1.09 | 0.86 |
| Carbapenem | 0.34 | 1.12 | 0.76 |
| Metronidazole | 1.11 | 0.95 | 0.24 |
| Vancomycin (intravenous) | 1.44 | 0.95 | 0.13 |

*Hosmer and Lemeshow Goodness of fit test p = 0.7536.

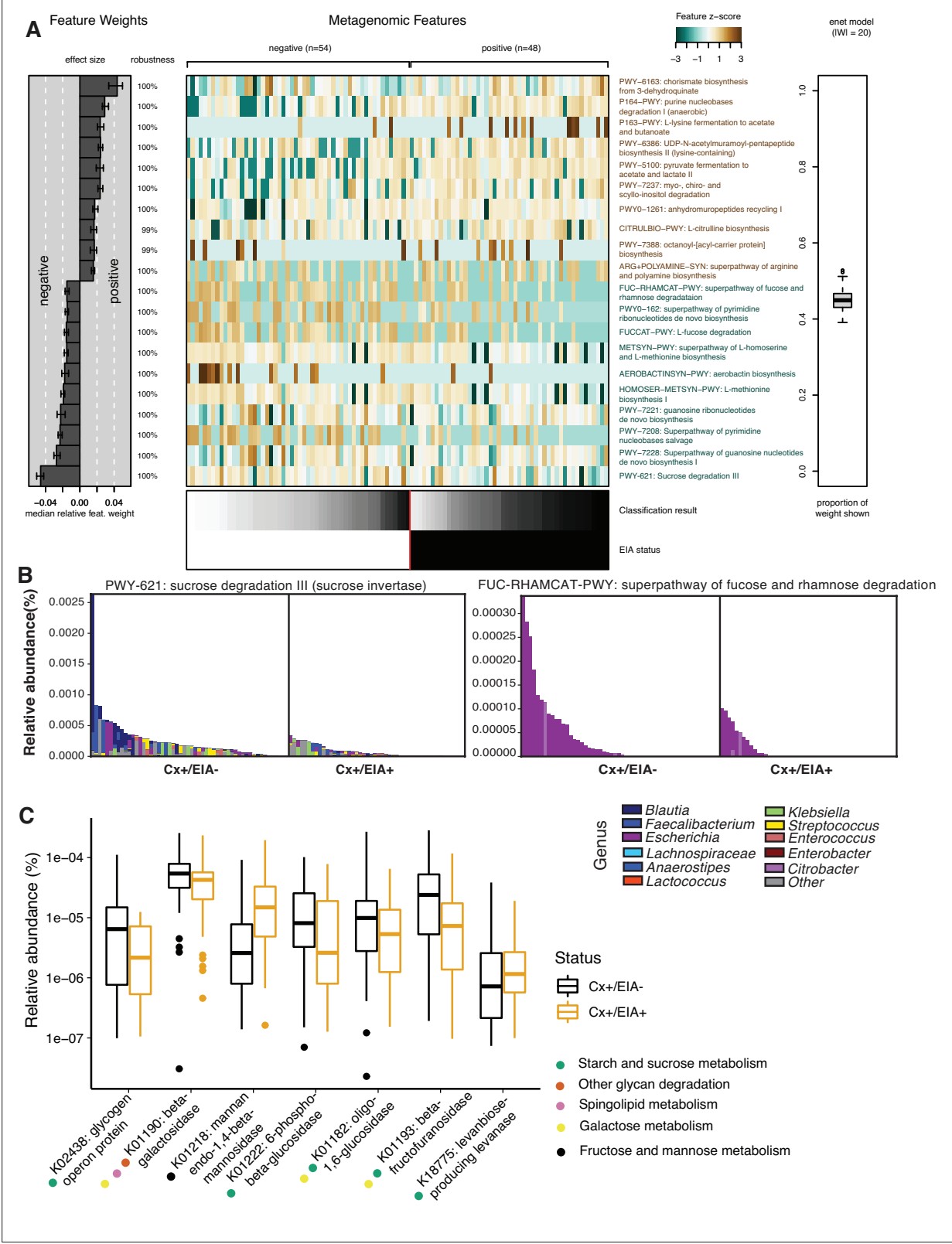

**Figure 2.** Carbohydrate metabolic processes present in asymptomatic patient microbiome. (**A**) Significant pathways associated with EIA status, as derived from the elastic net model. Mean-prediction AUC for the elastic net model was 0.825. (**B**) Relative abundance of taxa in pathways associated with asymptomatic patients, where each patient's metagenomic relative abundance is depicted by a single bar. Bars are colored by genera predicted to encode pathway. (**C**) Relative abundance of glycosidic hydrolase genes significantly associated with EIA status (q-value of <0.25) in stool metagenomes,

*Figure 2 continued on next page*

*Figure 2 continued*

where circles represent KEGG pathway classification.

The online version of this article includes the following figure supplement(s) for figure 2:

**Figure supplement 1.** Compositional measurements of metabolic pathways and metabolites.

associated with the asymptomatically-colonized (Cx+/EIA-) patients, including sucrose degradation III and fucose and rhamnose degradation. Investigation of the genera that encode such pathways revealed that the sucrose degradation III pathway was increased in asymptomatic patients, largely due to *Blautia* spp. and *Faecalibacterium* spp. of the class Clostridia (*Figure 2B*). Interestingly, the fucose and rhamnose degradation pathways were entirely defined by *Escherichia spp.*, presumably *E. coli*. This suggests that metabolic functions such as fucose and rhamnose degradation may be confined to a smaller number of taxa than carbohydrate degradation pathways such as sucrose degradation. Using the HUMAnN2 (*Franzosa et al., 2018*) gene family information, we used linear mixed modeling to identify carbohydrate-active enzymes differentially associated with EIA status (*Figure 2C*). Supporting the pathway analysis, we found an increased abundance of a subset of glycoside hydrolase genes, specifically involved in sucrose and starch metabolism in the asymptomatically

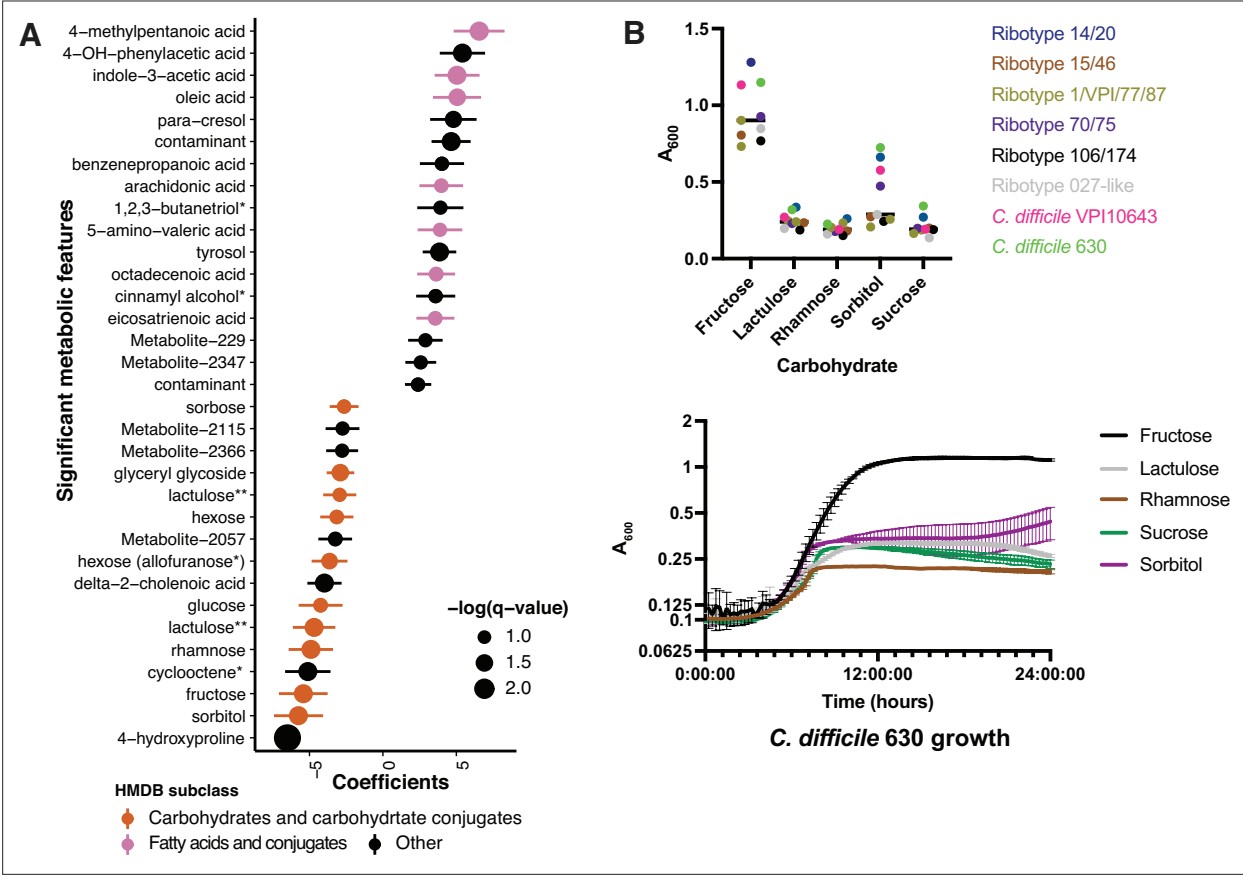

**Figure 3.** Asymptomatically colonized patients are defined by carbohydrate species. (**A**) Significant metabolites associated with EIA state, where a positive coefficient is strongly associated with Cx+/EIA+ metabolomes. Colors indicate human metabolic database (HMDB) sub-classification, and size of circle corresponds to magnitude of statistical significance. * indicates closest potential match; ** indicates two peaks from the same compound; contaminant indicates mass spectrometry contaminant. (**B**) Clinical and reference strains grown in *C. difficile* minimal medium (CDMM) with equimolar amounts of carbohydrate sources added. Growth was measured by taking maximum absorbance values over 48 hr. Each point represents the mean of two technical replicates of a unique isolate (top). Growth of *C. difficile* 630, from same conditions as above (bottom).

The online version of this article includes the following source data and figure supplement(s) for figure 3:

**Source data 1.** Growth curve data for *C. difficile* isolates.

**Figure supplement 1.** Validation of significantly associated metabolites.

colonized patients. Our metabolic pathway analyses highlight differentially abundant carbohydrate degradation processes in clostridial taxa that could contribute to colonization resistance against *C. difficile* in patient microbiomes.

We hypothesized that differences in metabolic potential of fecal microbiome communities might be reflected in metabolomic profiles, and therefore sought to identify metabolites that are altered in CDI patients relative to those asymptomatically colonized with *C. difficile* (*Robinson et al., 2019*). Ordination of Euclidean distances between Cx+/EIA- and Cx+/EIA+ stool metabolomes revealed no significant differences in metabolome structure (*Figure 2—figure supplement 1C*, PERMANOVA = 0.426). We again used MaAslin2 to determine metabolites associated with each disease state. Consistent with previous analysis, a number of end-product Stickland fermentation metabolites (4-methypentanoic acid and 5-aminovalerate) were associated with CDI patients. While 4-hydroxyproline was the strongest predictor of asymptomaticallycolonized patients, many of the significant metabolites that were associated with asymptomatic patients were predicted to be carbohydrates (*Figure 3A*, FDR < 0.25; *Supplementary file 3*). Putative metabolite identities were initially annotated by matching metabolite spectra to the NIST14 GC-MS spectral library. The preponderance of carbohydrates in asymptomatically colonized patients and the substantial similarity of carbohydrate spectra prompted us to rigorously validate the identities of these metabolites by comparing EI spectra and GC retention times against authentic standards, where commercially available (*Supplementary file 3*, *Figure 3—figure supplement 1*). These data reveal a carbohydrate signature that is depleted in CDI patients. Notably, fructose and rhamnose are either substrates or products of the sucrose degradation III and fucose and rhamnose degradation pathways, which we found to be enriched in asymptomatically colonized patients. The co-occurrence of these microbial pathways and their corresponding metabolites in asymptomatically colonized patients suggests that a commensal carbohydrate catabolism may contribute to suppression of *C. difficile* pathogenesis.

Our examination of taxa, metabolic pathways, and metabolites revealed a number of carbohydrates which we predict are undigestible by *C. difficile* or are end-products of a more complex commensal metabolism that is exclusionary to *C. difficile*. Using a set of clinical *C. difficile* isolates cultured from this patient cohort (8 isolates representing six different ribotypes), we examined growth of *C. difficile* on carbohydrates associated with asymptomatically colonized patients. Using a defined minimal media (CDMM)( *Karasawa et al., 1995*) to test nutrient utilization, we found that *C. difficile* isolates grew robustly on fructose as expected (median maximum $A_{600}$ of 0.90), but did not proliferate on rhamnose or lactulose (median maximum $A_{600}$ of 0.19 and 0.24, respectively). Notably, in the case of sorbitol, we found that a subset of strains, including the reference strain *C. difficile* 630 and *C. difficile* VPI10643, grew to a maximum $A_{600}$ of greater than 0.47 (*Figure 3B*). Given that we had found sucrose degradation as a metabolic pathway enriched in asymptomatically colonized patients, we hypothesized that *C. difficile* would be unable to use this carbohydrate. Indeed, when grown on sucrose as the sole carbon source, strains achieved a median maximum $A_{600}$ ~4.7-fold less than that of growth on fructose. *C. difficile's* restricted carbohydrate metabolism, coupled with the presence of commensal Clostridia could hamper progression to CDI.

We hypothesized that the differential abundance of identified stool metabolites in these patient cohorts is related to the metabolism of specific microbes or host processes. We performed a sparse partial least-squares-discriminatory analysis (sPLS-DA) with the mixOmics package to define relationships between the most predictive features of patient metabolomes and microbiomes. We optimized the number of latent components (*Figure 4—figure supplement 1A*) and number of variables (*Figure 4—figure supplement 1B*). Our final model contained two latent components, with the first one composed of 15 metabolites and 25 microbial species. Of the largest metagenomic variable weights, four out of five species (*C. difficile*, a *Lachnospiraceae spp.*, *Anaerostipes hadrus*, and *Clostridium clostridioforme*) were also significantly associated with an EIA state (*Figure 1*). Of the metabolomic variable weights (*Figure 4A*), the 10 highest-weighted metabolites were also discovered by previous analyses (*Figure 2*). The predictive value of each of the components per block was greater that an area under the curve (AUC) of 0.85, with the second metagenomic block component having the best performance (AUC = 0.94, *Figure 4—figure supplement 1C*). The strong performance of the latent components in classifying samples via EIA status validated our previous findings. Using the variables defining the first latent component, we performed correlational analyses (*Figure 4B*) and found a number of striking correlations. *C. difficile* abundance was positively correlated with a

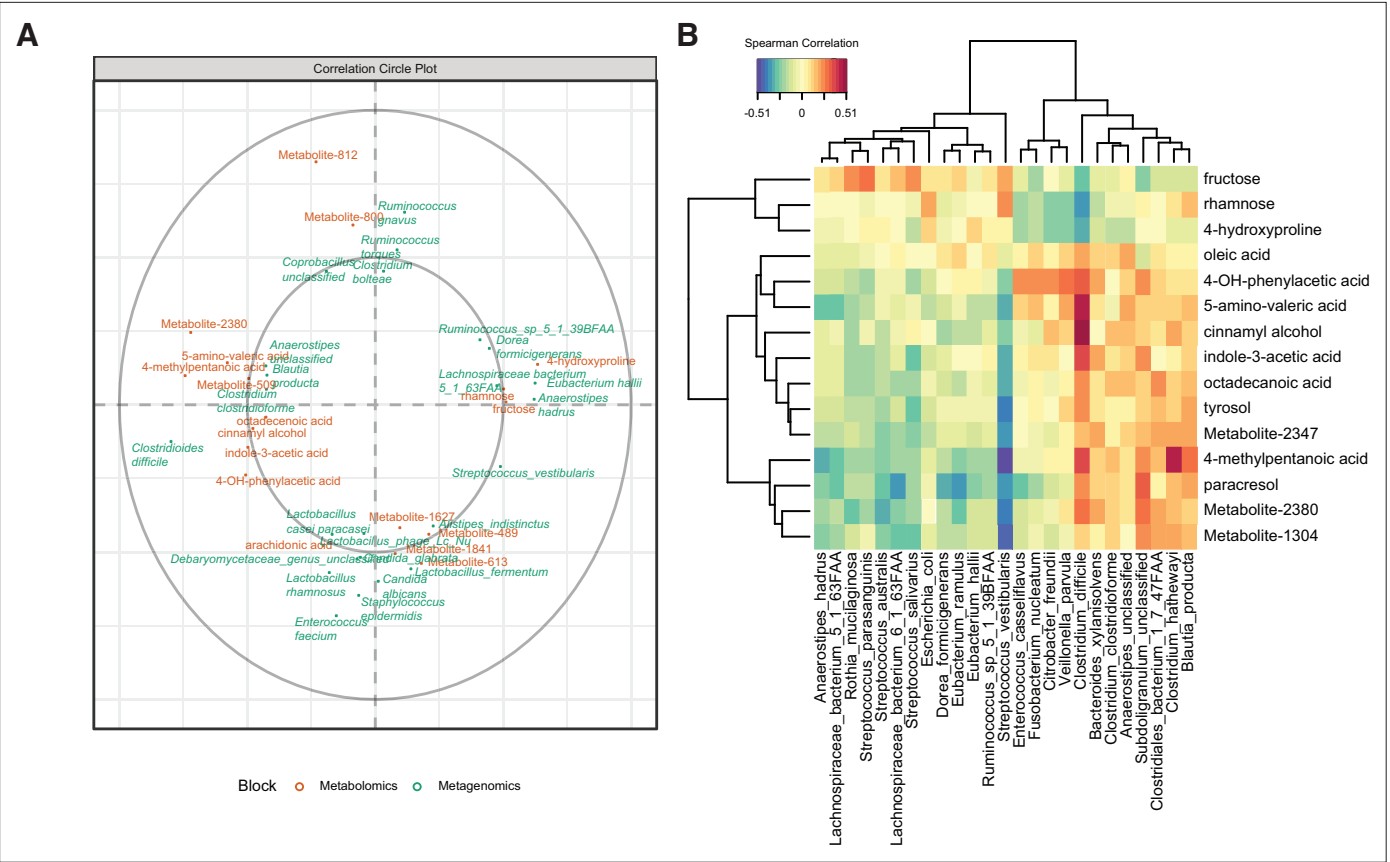

**Figure 4.** Multi-omics signature of *C. difficile*-colonized patients reveals *C.difficile*-metabolite relationships. (**A**) Correlation circle indicating the contribution of each variable (microbe or metabolite) to latent component of sparse partial least-squares-discriminatory analysis (sPLS-DA) using MetaPhlAn2 data. (**B**) Heatmap of Spearman correlations between metagenomic and metabolomic variables from the first latent component using MetaPhlAn2 data.

The online version of this article includes the following figure supplement(s) for figure 4:

**Figure supplement 1.** Multi-omics analysis performance.

number of well-known Stickland metabolites (5-amino-valeric acid and 4-methylpentanoic acid, rho = 0.48 and 0.36, respectively)(***Robinson et al., 2019***), whereas *C. difficile* had negative correlations with fructose, rhamnose, and hydroxyproline (rho = –0.27, 0.36, and –0.34, respectively). Given our metagenomic data suggesting that Kraken metagenomic profiling yielded more sensitive estimates of *C. difficile* abundance, we performed an independent multi-omics analysis on the same dataset using the Kraken metagenomic data. Using a similar process of model building as above, the final model consisted of two latent components, with 15 metabolites and 15 microbes in the first component (***Figure 4—figure supplement 1C***). *C. difficile* was also most positively correlated to 5-amino-valeric acid and 4-methylpentanoic acid, with corresponding negative correlations to fructose, rhamnose, and hydroxyproline (***Figure 4—figure supplement 1D***). These microbe-metabolite relationships highlight the known pathophysiology of CDI, and identify novel *C. difficile*-carbohydrate relationships that define asymptomatic colonization.

Given the anticorrelation between *C. difficile* and rhamnose, we sought to explain the enrichment of this carbohydrate in asymptomatically colonized patients. Though *C. difficile* cannot grow on rhamnose as the sole carbohydrate, in other organisms rhamnose has substantial transcriptional influence over carbon catabolite gene clusters (***Egan and Schleif, 1993***; ***Hirooka et al., 2015***). We wanted to rule out the possibility that rhamnose may impact *C. difficile* through possibly cryptic transcriptional reprogramming, perhaps contributing to *C. difficile* repression in vivo. Accordingly, we performed whole transcriptome RNA sequencing on *C. difficile* cultures exposed to a metabolizable substrate, fructose, or a non-metabolizable substrate, rhamnose. In the presence of fructose, we found 555

genes significantly altered (adjusted p-value < 0.05 and |fold-change| > 2) (**Supplementary file 4**). Some of the most altered genes were indicative of carbon catabolite repression of sugar transport and upregulation of glycolytic processes to metabolize fructose. In contrast, we found only three genes significantly increased in the rhamnose condition. The lack of striking systems-level or targeted (toxin expression, sporulation) regulation by rhamnose, and *C. difficile*'s inability to utilize it, leads us to conclude that its association with asymptomatically colonized patients' microbiomes is not through direct interaction or suppression of *C. difficile*. Instead, we speculate that rhamnose may be the byproduct of a complex commensal metabolism of other dietary polysaccharide substrates, which could exclude *C. difficile* from the GI tract.

## Discussion

### Factors affecting the outcome of *C. difficile* colonization

Susceptibility to CDI is the result of a complex interaction between host factors (variation in bile acid metabolism, adaptive immunity) and abiotic factors such as antibiotic treatment and diet (**Mullish and Allegretti, 2021**; **Littmann et al., 2021**). These variables largely affect colonization resistance in the gut microbiome community and influence pathogen proliferation (germination rate, variation in toxin activity, and metabolic capacity). Our study endeavored to identify gut microbiome signatures (both taxonomic and metabolic), bacteriologic traits, and antibiotic exposure histories that might help explain Cx+/EIA- *C. difficile* colonization. Clinically, this manifestation is an intermediate state on the spectrum of *C. difficile*- associated disease and correspondingly, a diagnostic conundrum. One limitation of this study is our inability to assess dietary histories of patients leading up the diagnostic event. The metabolomic data provides a snapshot in time. While we hypothesize the increase in monosaccharides is due to an increase in carbohydrate degradation within the community, it is unclear whether the carbohydrate signature is due to microbial community differences in cross-feeding rates or differences in host diet.

Another important limitation to this study is our inability to control for *C. difficile* strain differences and correspondingly, heterogeneity in processes such as spore germination, nutrient utilization, and toxin expression in vivo (**Kumar et al., 2019**; **Hunt and Ballard, 2013**). Strains infecting Cx+/EIA+ patients were more likely to contain the *cdtAB* toxin locus, and the distribution of ribotypes was qualitatively different between the two cohorts (indicating significant pathogen variation). In vitro examination of toxin production (TcdA and TcdB) using a commercial ELISA indicated that over half of isolates expressed detectable levels of toxins. Toxin expression is well-known to be regulated by nutrient conditions and although our in vitro data indicate that both cohorts contain similar numbers of strains capable of producing toxin in vitro, such conditions are considered inadequate to predict in vivo levels of toxin production (**Burnham and Carroll, 2013**; **Akerlund et al., 2006**). Further, we found that a diverse set of clinical *C. difficile* strains might have variation in their ability to utilize nutrients such as sorbitol, which contrasts with reports of model *C. difficile* strains harboring more flexibility in their ability to utilize nutrients (**Theriot et al., 2014**; **Jenior et al., 2017**; **Scaria et al., 2014**). The outcome of strain level differences in metabolism and virulence is further complexified by in vivo conditions that might influence pathogen proliferation. Yet, we speculate that certain gastrointestinal environments both encourage some growth of *C. difficile* and discourage the elaboration of toxin, as toxin expression is actively repressed in nutrient-rich conditions.

Antibiotic treatment is the most well-understood risk factor for CDI (**Stevens et al., 2011**; **Deshpande et al., 2013**), and antibiotic exposure in our cohort likely results in loss of the species we find depleted from CDI patients. Here, we confirm that exposure to a number of antibiotics is associated with CDI patients, including cephalosporins (significantly associated) and intravenous vancomycin (weakly associated). Clindamycin and quinolones, two antibiotics also associated with CDI in other human cohorts (**Teng et al., 2019**) are likely not significantly associated in our population due to the low prevalence of their exposure. Our microbiome data reveals decreased levels of *Streptococcus, Ruminococcus, and Eubacterium* spp. in CDI patients. Findings from both human cohorts and mouse models of antibiotic treatment indicate that a number of clostridial taxa are depleted upon administration of a variety of antibiotic treatments (**Palleja et al., 2018**; **Rashid et al., 2015**). It is also posited that some of these taxa are integral to protection from CDI (**Mills et al., 2018**). Given the attempts to use FMTs or Firmicutes-enriched probiotics to prevent CDI, we hypothesize that the restoration of

lost species from class Clostridia after high-risk antibiotic treatment could be a novel avenue for CDI prevention (*McGovern et al., 2021*).

## Gut metabolites as markers of *C. difficile* proliferation and the microbiome

While metabolites associated with CDI and correlated with *C. difficile* abundance (4-methyl-pentanoic acid and 5-amino-valeric acid) clearly reflect *C. difficile* proliferation (*Akerlund et al., 2006*), the metabolites associated with Cx+/EIA- patients could reflect a number of non-mutually exclusive biological scenarios, indicating either the absence of *C. difficile* proliferation or the presence of a stable community where *C. difficile* pathogenesis is prevented by community metabolic elements.

In the one scenario, we reference two metabolites, 4-hydroxyproline and sorbitol, which have been considered host products of collagen degradation and inflammation (*Fletcher et al., 2021*; *Pruss and Sonnenburg, 2021*). The abundance of 4-hydroxyproline in the stools of Cx+/EIA- and its anti-correlation with *C. difficile* levels would suggest that it is a substrate consumed by *C. difficile* during pathogenesis. In a mouse model of CDI, sugar alcohols and amino acids observed before infection were considered representative of a 'pre-colonized state' (*Fletcher et al., 2018*; *Theriot et al., 2014*), as these nutrients declined as CDI progressed. However, we restricted our cohort to patients who were not on their way to developing CDI, by excluding patients with EIA- stool if they were subsequently diagnosed with CDI or received empiric CDI treatment within 10 days of initial stool collection (*Dubberke et al., 2018*).

In another scenario, the overlap of signatures between pathways, metabolites, and microbes highlights a number of possible metabolic pathways that might be exclusionary to *C. difficile,* namely starch/sucrose degradation and rhamnose degradation. The combination of our microbiome data, which shows enrichment of number of commensal Clostridia such as *Eubacterium* spp.(*Desai et al., 2016*), starch/sucrose degradation pathways, and our in vitro data highlights a possible microbe-metabolite combination that could prevent *C. difficile* proliferation. Rhamnose is a major component of plant and some bacterial cell-wall polysaccharides (*Silva et al., 2020*). Metabolic pathway profiling revealed an enrichment of fucose and rhamnose degradation pathways in asymptomatically colonized patients, represented by Enterobacterales taxa. Therefore, we propose that the detected rhamnose is a byproduct of commensal catabolism of more complex polysaccharides containing rhamnose (*Porter and Martens, 2017*; *Mistou et al., 2016*). These findings are of course limited by the scope of the in vitro experiment and the correlative nature of our microbiome data. Future work examining in vivo competition between diverse *C. difficile* isolates and commensal isolates with critical metabolic elements would be required.

Lactulose was a carbohydrate associated with asymptomatically-colonized patients and not a robust growth substrate for *C. difficile*. Interestingly, lactulose has been previously associated with a decrease in *C. difficile*-related diarrhea (*Maltz et al., 2020*) and decreased risk of CDI (*Maltz et al., 2020*; *Agarwalla et al., 2017*). Lactulose is a disaccharide product from heat treatment of lactose (a common sugar in dairy products), but it is also a component of some laxatives (*Adachi, 1958*). However, patients were screened and excluded from this cohort if they were prescribed laxatives in the 24 hr prior to sample collection. In addition to this screening/exclusion criteria, lactulose is almost exclusively prescribed to liver failure patients (there were none reported in this study), thus it is more likely to be present from consumption of heated milk (containing lactose). Other in vitro work demonstrates that addition of 'non-digestible' oligosaccharides, such as lactulose, provides a competitive advantage to *Bifidobacterium* spp. over *C. difficile* (*Kondepudi et al., 2012*; *Hopkins and Macfarlane, 2003*). While we do not recommend lactulose, a laxative, as such a prebiotic, there are a number of other 'non-digestible' oligosaccharides that might serve similar purposes in future interventions (*Hopkins and Macfarlane, 2003*). Taken together, these data emphasize the potential for synthetic or natural prebiotic interventions to shift a vulnerable microbiota away from CDI.

## Strategies to ameliorate toxigenic *C. difficile* proliferation

Our multi-omics analyses of a colonized asymptomatic patient population support a growing body of literature concerning commensal metabolism as a tool against *C. difficile*. Evidence from both mouse models of disease and human studies indicate that administration of polysaccharides or 'microbial accessible carbohydrates' may prevent *C. difficile* proliferation or decrease the risk of CDI (*Mefferd*

*et al., 2020*; *Schnizlein et al., 2020*; *Hryckowian et al., 2018*; *Maltz et al., 2020*; *Lewis et al., 2005*). Recently, a probiotics-based attempt to design a consortium of mucosal sugar utilizers revealed its ability to decrease *C. difficile* colonization in vivo (*Pereira et al., 2020*), indicating that increasing mucosal metabolism, or carbohydrate catabolism, may be another route to strengthening commensal resistance to *C. difficile.* Interestingly, previous attempts to combinatorically assemble species and nutrient combinations that might inhibit *C. difficile* indicate that success is afforded by species able to competitively utilize carbohydrates such as sorbitol and mannitol (*Ghimire et al., 2020*). Given the plethora of prebiotics and probiotics explored in the *C. difficile* field, we emphasize the need for an approach that harnesses both probiotic- and prebiotic-based components to inhibit the proliferation of *C. difficile* and toxin-mediated pathogenesis.

## Materials and methods
### Patient cohort analysis

A previous retrospective cohort study was conducted to understand *C. difficile* colonization. In that study, *C. difficile* isolates were cultured from patient stool as described. Ribotyping was performed using the DiversiLab Bacterial Barcodes software (bioMerieux) (*Dubberke et al., 2018*; *Westblade et al., 2013*). Analysis of isolate genetic traits and in vitro toxin production was performed on the 102 isolates for which we had corresponding metagenomic sequencing data (see below). Data concerning isolate ribotype was aggregated into the three most abundant ribotypes (ribotype 027, ribotype 106, ribotype 14/20), where all other ribotypes or unclassified strains were grouped into 'Other'. For the purposes of this study, data concerning inpatient antibiotic orders were retrospectively collected from the electronic medical informatics database for patients with toxin EIA positive (Cx+/EIA+) stool (n = 62) or toxin EIA negative (Cx+/EIA-) stool (n = 62). The presence of antibiotic orders was classified into three dichotomous groups by timing of exposure: antibiotics in 0–7 days before stool collection (1 week), antibiotics in >7–14 days before stool collection (2 weeks), and antibiotics in >14–30 days before stool collection (1 month). To understand the specific antibiotics associated with EIA status in our patient cohort, raw antibiotic exposure data was aggregated by time. Additionally, low-prevalent antibiotics ( < 10% exposure in patients) were removed from analysis. Logistic regression analysis was performed using the *glm* function in R. To understand overall antibiotic exposure as it relates to EIA status, any antibiotic exposure was considered '1' and zero antibiotic exposure in a patient was considered '0'. The binary antibiotic exposure variable was then used in linear mixed modeling analysis to understand species associated with antibiotic exposure.

### Metagenomic sequencing and analysis of patient stool

Metagenomic DNA was extracted from patient stools as previously described (*Fishbein et al., 2021a*). *C. difficile* was isolated from patient stools as previously described (*Fishbein et al., 2021a*). Illumina libraries of patient stool metagenomic DNA were prepared and pooled as previous described (*Fishbein et al., 2021a*; *Baym et al., 2015*). Fecal metagenomic libraries were submitted for 2 × 150 bp paired-end sequencing on an Illumina NextSeq High-Output platform. Reads were binned by index sequences and reads were trimmed and quality filtered using Trimmomatic v.0.38 (*Bolger et al., 2014*) to remove adapter sequences and DeconSeq (*Schmieder and Edwards, 2011*) to remove human sequences. Samples that were less than 15% bacterial DNA during initial sequencing were discarded, and all samples were sequenced to a depth of at least 5 million reads. Sample loss due to low bacterial DNA resulted in a smaller cohort than originally reported (*Dubberke et al., 2018*), with the final set of metagenomes representing 54 Cx+/EIA- and 48 Cx+/EIA+ patients.

We performed taxonomic profiling of metagenomic sequences using MetaPhlAn2 (*Truong et al., 2015*), and functional pathway profiling using HUMAnN2 (*Franzosa et al., 2018*). MetaCyc pathway abundances were normalized to relative abundances using the *humann2_renorm.py* function. The *humann2_barplot.py* function was used to assess taxonomic composition of metabolic pathways. Custom python scripts were used to parse MetaPhlAn2 '_profiled_metagenome.txt' and HUMAnN2 'pathwayabundance.txt' files. Data were imported to R to analyze community composition and differential associations. To analyze carbohydrate-active enzymes, we used *humann2_regroup.py* and *humann2_rename.py* function to reannotate the '_genefamilies.txt' files and identify genes with

the enzyme classification number EC:3.2.1.*, representing glycosidases, enzymes that participate in carbohydrate degradation (*Ghimire et al., 2020*).

## Metagenomic data analysis

For both microbiome and metabolomic data, the *nearZeroVar* function of the caret package was used to remove low-prevalent or invariant taxa/pathways/metabolites (*Kuhn, 2008*). These filtered data sets were analyzed for differential association and multi-omics modeling. Alpha-diversity and beta-diversity were calculated using the vegan package. Weighted UniFrac distance was used as a beta-diversity metric for microbial taxa and Bray Curtis dissimiliarity was used as a beta-diversity metric for metabolic pathways, while Euclidean distance was used as a beta-diversity metric for metabolomes. The MaAslin2 package was used for linear mixed modeling to identify microbial taxa, gene families, and metabolites associated with EIA/antibiotic exposure status.

To analyze HUMAnN2 pathways enriched in cohorts, we used statistical inference of associations between microbial communities and host phenotypes (SIAMCAT) (*Wirbel et al., 2021*), using the siamcat package in R, to fit an elastic net model to the data. We used the following parameters: log. std normalization, 10 folds and 10 resamples for data splitting. The *model.interpretation.plot* function was used to display features weights for features used in >70% of models generated in training.

## Determination of candidate metabolites

Putative identification of metabolites of interest (*Supplementary file 3*) was initially performed through spectral matching against the NIST14 electron ionization spectrum library. Several features were previously identified by our group (see *Robinson et al., 2019*). Features predicted to be sugars or sugar alcohols were compared to a panel of authentic standards (D-sorbitol, D-mannitol, D-fructose, L-rhamnose, L-fucose, lactulose, glucose, mannose, D-galactose, D-talose, *myo*-inositol, and L-sorbose). Because isomeric sugars generate very similar spectra, we utilized both spectral similarity and retention time to identify sugar metabolites (*Figure 3—figure supplement 1*).

## Multi-omics analysis

The metagenomic relative abundance data was imputed with min(abundance >0)/2, and the metabolomic data was imputed with a value of 1. For both filtered datasets, a centered log-ratio transformation was used to analyze filtered metagenomic and metabolomic datasets above. The mixOmics (*Rohart et al., 2017*) package in R was used for multi-omics analysis of both MetaPhlAn and Kraken metagenomic relative abundance data. To avoid over-fitting on the large number of variables in our datasets, we utilized sPLS-DA. Briefly, to determine the number of variables from each dataset to keep in the final model, we estimated model error rates for all combinations of seq(15,30,5) variables for both metagenomic and metabolomic datasets, using the function tune.block.splsda (10-fold cross-validation, repeated 50 times, "max.dist" distance metric). Spearman correlations were calculated between CLR-transformed microbial taxa and metabolite abundances, from the variables defining the first latent components, and plotted using the cim package.

## Bacteriology and in vitro growth assays

*C. difficile* strains were isolated from patient stools by plating on cycloserine-cefoxitin fructose agar as previously described; strains were stored at –80°C (*Fishbein et al., 2021a*). *C. difficile* VPI10643 and *C. difficile* 630 reference strains were purchased from ATCC, and included in the assays described below using the same conditions as clinical isolates. For in vitro growth assays, CDMM was prepared as previously described (*Karasawa et al., 1995*) and 20 mM of specified carbohydrates were added. Clinical isolates were inoculated into tryptone-yeast extract (TY) broth and grown for 16 hr, then washed with PBS and diluted 1:100 into media with different carbohydrates sources. Growth was measured in a shaking, 96-well plate at 37°C for 48 hours.

In vitro ELISAs to assess toxin production in each isolate were performed on using TGCbiomics kits for 'Simultaneous detection of TcdA and TcdB' and 'C. difficile GDH detection kit' as a control ELISA. Cultures were grown for 24 hr in TY media in deep 96-well plate. Following, cultures were spun down and culture supernatants were diluted 1:5 in dilution buffer and loaded onto ELISA plates for detection of both toxin and control protein (GmbH), per manufacturer's instructions. Isolates were considered positive for toxin if they had greater absorbance than that of the negative control.

## RNA sequencing and data analysis

Five mL of each strain (in biological triplicate) were grown to log-phase ($OD_{600}$ ~0.4)in TY and exposed to TY- rhamnose or TY-fructose (with each carbohydrate at 30 mM). Cells were harvested by adding one volume of 1:1(v/v) acetone/ethanol to the culture to arrest growth and RNA degradation. Sample were spun at 4000 x g for 5 min. The cell pellet was washed with 500 μl TE buffer (0.5 M EDTA, 1 M Tris pH 7.4) and spun down to remove the supernatant. The cell pellet was resuspend in one mL Trizol and two rounds of bead-beating at 4500 rpm for 45 s were performed. A total of 300 μl of chloroform was added to the suspension, lysates were vortexed, and centrifuged at 4000 rpm for 10 min at 4°C. The aqueous layer was removed and RNA was precipitated using isopropanol, washed with 70% ethanol, and resolubilized in TE buffer. Total RNA was treated with Turbo DNase (for two rounds of digestion). rRNA depletion was performed using the QiaFast-Select kit (Hilden, Germany), following manufacturer's instructions. Libraries were prepared using the rRNA-depleted RNA as input for NEBNext Ultra II RNA Library Prep Kit (NEB, Ipswich, MA). Libraries were pooled and submitted for 2 × 150 bp paired-end sequencing on an Illumina NextSeq High-Output platform at the Center for Genome Sciences and Systems Biology at Washington University in St. Louis.

Raw reads were trimmed using Trimmomatic v. 0.38, and aligned to a *C. difficile* VPI10643 reference genome (GCF_000155025.1) using Bowtie2. SAM files were converted to BAM format and indexed using samtools. Read counts for each gene feature were obtained using the featureCounts function of subread-1.6.5 package. Counts were manually imported into R, and DEseq2 was used to identify differentially expressed gene products in the case of TY-fructose relative to TY and TY-rhamnose relative to TY.

## Data deposition

Metagenomic reads were deposited under BioProject accession number PRJNA748262 and RNA sequencing reads were deposited under BioProject accession number PRJNA748261. All R code and metadata used to generate figures is deposited at https://github.com/srsfishbein/2021EIACdiff_multiomics, (*Fishbein, 2021b* copy archived at swh:1:rev:0c2a33d873e43194afb5818733e46c6ff28d6947).

## Acknowledgements

The authors are grateful for members of the Dantas lab for their helpful feedback on the data analysis and preparation of the manuscript. The authors are specifically grateful to Drew J Schwartz for his insightful feedback. The authors would also like to thank the Edison Family Center for Genome Sciences and Systems Biology staff, Eric Martin, Brian Koebbe, MariaLynn Crosby, and Jessica Hoisington-López for their expertise and support in in sequencing/data analysis.

This work was supported in part by awards to ERD from the CDC Broad Agency Announcement, contract 200-2017-96178). JPH was supported by CDC (Broad Agency Announcement, contract (Broad Agency Announcement, contract 200-2019-05950) and the National Institute of Diabetes, Digestive, and Kidney Diseases of the National Institutes of Health (RO1DK111930). GD received support from the National Center for Complementary and Integrative Health (NCCIH: https://nccih.nih.gov/) of the National Institute of Health (NIH) under award number R01AT009741; the National Institute for Occupational Safety and Health (NIOSH: https://www.cdc.gov/niosh/index.htm) of the US Center for Disease Control and Prevention (CDC) under award number R01OH011578l, and the Congressionally Directed Medical Research Program (CDMRP: https://cdmrp.army.mil/prmrp/default) of the US Department of Defense DOD under award number W81XWH1810225. SRSF is supported by the T32 Pediatric Gastroenterology Research Training Program under the National Institute of Child Health and Human Development (NICHD: https://www.nicdhd.nih/gov) of the NIH under award number T32DK077653 (PI: P.I. Tarr).

## Additional information

### Competing interests

Erik R Dubberke: E.R.D. has received research support from Synthetic Biologics, Pfizer and Ferring, and has been a consultant for Summit, Merck, Ferring, Pfizer and Seres Therapeutics, all unrelated to this study. The other authors declare that no competing interests exist.

### Funding

| Funder | Grant reference number | Author |
|---|---|---|
| National Institute of Diabetes and Digestive and Kidney Diseases | RO1DK111930 | Jeffrey P Henderson |
| Centers for Disease Control and Prevention | 200-2017-96178 | Erik R Dubberke |
| National Center for Complementary and Integrative Health | R01AT009741 | Gautam Dantas |
| National Institute for Occupational Safety and Health | R01OH011578l | Gautam Dantas |
| Congressionally Directed Medical Research Programs | W81XWH1810225 | Gautam Dantas |
| Eunice Kennedy Shriver National Institute of Child Health and Human Development | T32 HD004010 | Skye RS Fishbein |

The funders had no role in study design, data collection and interpretation, or the decision to submit the work for publication.

### Author contributions

Skye RS Fishbein, Conceptualization, Formal analysis, Investigation, Methodology, Validation, Visualization, Writing – original draft, Writing – review and editing; John I Robinson, Formal analysis, Investigation, Methodology, Validation, Visualization, Writing – review and editing; Tiffany Hink, Data curation, Methodology; Kimberly A Reske, Data curation, Methodology, Project administration, Writing – review and editing; Erin P Newcomer, Investigation, Writing – review and editing; Carey-Ann D Burnham, Data curation, Investigation, Supervision, Writing – review and editing; Jeffrey P Henderson, Conceptualization, Funding acquisition, Investigation, Methodology, Supervision, Writing – review and editing; Erik R Dubberke, Conceptualization, Data curation, Formal analysis, Funding acquisition, Investigation, Resources, Supervision, Writing – review and editing; Gautam Dantas, Conceptualization, Funding acquisition, Investigation, Methodology, Project administration, Supervision, Validation, Visualization, Writing – review and editing

### Author ORCIDs

Skye RS Fishbein (i) http://orcid.org/0000-0002-9554-1170
John I Robinson (i) http://orcid.org/0000-0003-3107-0047
Gautam Dantas (i) http://orcid.org/0000-0003-0455-8370

### Decision letter and Author response

Decision letter https://doi.org/10.7554/eLife.72801.sa1
Author response https://doi.org/10.7554/eLife.72801.sa2

---

## Additional files

### Supplementary files

- Supplementary file 1. Patient demographic data.
- Supplementary file 2. Fecal metagenomics metadata file with isolate information.
- Supplementary file 3. MaAsLin2 output of metabolites associated with EIA status in addition to metabolite validation information.

• Supplementary file 4. DEseq output of in vitro rhamnose-exposed *C difficile* transcriptomic profiling.

• Transparent reporting form

## Data availability

Metagenomics reads were deposited under BioProject accession number PRJNA748262 and RNA sequencing reads were deposited under BioProject accession number PRJNA748261.

The following dataset was generated:

| Author(s) | Year | Dataset title | Dataset URL | Database and Identifier |
|-----------|------|---------------|-------------|------------------------|
| Fishbein SRS | 2021 | Fecal metagenomes of C. difficile colonized patients | https://www.ncbi.nlm.nih.gov/bioproject/PRJNA748262 | NCBI BioProject, PRJNA748262 |
| Fishbein SRS | 2021 | C. difficile carbohydrate transcriptomics | https://www.ncbi.nlm.nih.gov/bioproject/PRJNA748261 | NCBI BioProject, PRJNA748261 |

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
