## [Editor Report]

Not everyone colonized by *C. difficile* has gut symptoms, but the reasons why are unclear. This article uses the combination of sequencing and mass spectrometry to compare patients with or without symptoms, revealing links between specific gut bacteria and diet, which could lead to diet or bacterial treatment or prevention strategies.

---

## [Decision Letter]

**Decision letter after peer review:**

Thank you for submitting your article "Multi-omics investigation of *Clostridioides difficile*-colonized patients reveals protective commensal carbohydrate metabolism" for consideration by *eLife*. Your article has been reviewed by 3 peer reviewers, including Peter J Turnbaugh as the Reviewing Editor and Reviewer #1, and the evaluation has been overseen by Gisela Storz as the Senior Editor.

Essential revisions:

1) Need to discuss the ability to exclude alternative hypotheses, including variations between C. difficile strains, dietary intake, differences in host physiology, and bile acid production/metabolism. The former seems like a critical point – are these individuals colonized by similar strains of C. difficile? Are they all toxin positive? It is critical to test if C. difficile from the Cx+/EIA- samples are actually capable of producing toxin. This is important to discern whether there are facets of the microbiome/metabolome which turn toxin off in Cx+/EIA- samples or if C. difficile in these patients have mutations which make them unable to produce toxin.

2) Given the compositional nature of the sequencing data it is possible that differences in C. difficile are responsible for some of the observed differences in community structure. Please mask C. difficile reads and re-run the key analyses to check if they hold up.

3) Please discuss the literature precedent for C. difficile growth on different carbohydrates and ideally include data for the type strain.

4) Please check if the conclusions are impacted by removing the Cx+/EIA- samples with metagenomically undetectable C. difficile from the computational analyses used in Figures 1-3. The concern is whether these samples are driving the perceived differences between Cx+/EIA+ patients and Cx+/EIA- patients (does C. difficile abundance or metabolite abundance still differentiate Cx+/EIA+ patients from Cx+/EIA- patients?).

*Reviewer #1:*

The mechanisms that protect some individuals from C. difficile-associated colitis remain poorly understood; however, recent data has implicated both diet and the microbiome. Here, the authors use paired metagenomic and metabolomic analysis to identify differences in asymptomatic and symptomatic patients, suggesting that competition between clostridial species for carbohydrate metabolism may play a role. The data is clearly presented and provides clear hypotheses for future studies aimed at understanding the complex interactions between enteric pathogens, the gut microbiome, and host pathophysiology.

Strengths of this study include its unique cohort, rigorous analysis and presentation, inclusion of some initial in vitro validation work, and potential for inspiring future hypothesis-driven experiments.

Weaknesses include the lack of consideration or ability to control for alternative hypotheses, including variations between C. difficile strains, dietary intake, differences in host physiology, and bile acid production/metabolism. The effect sizes are also modest with no high-level differences in the microbiome or metabolome between groups. Finally, there is no evidence of generalizability to other patient cohorts. Given these caveats, it is important to be clear throughout that this is a hypothesis generating exercise and that the degree to which commensal carbohydrate metabolism is protective against C. difficile infection requires further clinical and mechanistic data.

Comments for the authors:

1. Need to discuss the ability to exclude alternative hypotheses, including variations between C. difficile strains, dietary intake, differences in host physiology, and bile acid production/metabolism. The former seems like a critical point – are these individuals colonized by similar strains of C. difficile? Are they all toxin positive? I was unclear how asymptomatic carriage is defined, this is critical to the current paper and should be included in the main text and methods, not as a citation.

2. Given the compositional nature of the sequencing data it is possible that differences in C. difficile are responsible for some of the observed differences in community structure. I'd recommend masking C. difficile reads and re-running the key analyses to check if they hold up.

3. The in vitro validation is helpful, but I'm unclear as to whether it is new information. If any prior studies have been done they should be cited here.

*Reviewer #2:*

The manuscript by Fishbein et al. examines an exciting and timely question about the microbial and metabolomic factors in the gastrointestinal tract that determine if C. difficile remains dormant as a colonizer or triggers infection. They do this through the re-analysis of their previously published cohort wherein they can separate the colonization vs infection state on the basis of qPCR and toxin detection (EIA) which provides a unique opportunity to address their question in an excellently phenotyped cohort.

Using appropriate and current approaches, the authors find relatively subtle differences in the microbiome and metabolome of colonized/infected participants including certain carbohydrates which are elevated in asymptomatic individuals. They demonstrate these carbohydrates are not substrates for C. difficile; however, this line of experimentation sought to find negative results and it remains to be determined if they have any relevance in vivo or in complex communities. The authors could consider an additional experiment to probe this observation in more depth such as ex vivo fecal incubations or strain-strain competition experiments to provide more direct evidence for how they may influence the suppression of C. difficile.

Comments for the authors:

Given that this manuscript is primarily computational, it would be beneficial if the code for the analysis was shared in a public repository.

PRJNA748262 does not appear to be publicly available.

Line 66: would asymptomatic colonization be on the disease spectrum?

*Reviewer #3:*

This manuscript reveals exciting details of the lifestyle of C. difficile in individuals who are asymptomatic carriers and contrasts these with individuals with active CDI. It will be an important contribution to the C. difficile research field and more broadly to people interested in targeted manipulation of microbial communities.

The study team leveraged a unique patient cohort, coupled with metagenomic and metabolomic analyses, and bacterial culture techniques to suggest that the gut microbiome (namely, commensal clostridia) inhibit C. difficile proliferation. Much of the paper relies on the idea that C. difficile burdens are lower in asymptomatically colonized individuals versus those with C. difficile infection (CDI). However, many individuals who are asymptomatically colonized with C. difficile have burdens that are comparable to those who have CDI (and there may be a bimodal distribution of asymptomatically colonized individuals – those that have C. difficile detectable by metagenomic sequencing and those that do not). I worry that their conclusions are skewed by an apparent binary distribution of C. difficile burdens in both patient groups.

The patients are binned based on initial C. difficile diagnostic tests (asymptomatic individuals carry toxin-encoding C. difficile as detected by PCR but have no detectable toxin via immunoassay). I am unsure how sensitive the standard diagnostic immunoassay for C. difficile is but do know C. difficile toxin production is controlled by a complex regulatory network and abundance of these toxins can change with respect to parameters such as nutrient availability, temperature, pH, and cell density. I wonder if the isolates from the study are all capable of producing functional toxin. These strains could have mutations (in regulatory elements, for example) that may impact toxin production.

The authors perform a culture-based experiment to support their findings that certain metabolites present in the gastrointestinal tracts of asymptomatic individuals do not support the growth of C. difficile. They use isolates from the study participants but it is unclear if these isolates were taken from asymptomatic individuals or those with CDI but do not leverage strains commonly used in the research community (C. difficile 630 and C. difficile R20291).

Lines 67-69. What is the limit of detection of EIA? It's striking to me that, while C. difficile burdens are a predictor of Cx+/EIA+ vs. Cx+/EIA- patients, there are ~23 Cx+/EIA- patients whose relative abundance of C. difficile essentially superimposes with those of the Cx+/EIA+ group. There are fibroblast rounding assays that quantify C. difficile toxins. The limit of detection of these assays may be lower than kits used in diagnostic labs. Can the authors perform one of these assays on the samples in this study to better calibrate readers expectations of whether toxigenic C. difficile is or is not expressing toxin? As the paper stands, I assume that production of toxin in C. difficile is all-or-nothing, but prior work on regulation of toxin production in C. difficile has demonstrated that many regulatory cues influence toxin production (Martin-Verstraete et al. 2016) so I expect this to be non-binary. In addition, and considering the regulatory networks that control toxin production, are the C. difficile isolates from asymptomatic patients capable of producing toxin in vitro? Just because the gene is there doesn't mean that it's functional.

Line 140-142. "Given that C. difficile levels were an overt predictor of CDI…" Are C. difficile levels still a predictor of the Cx+/EIA+ state if the samples where C. difficile was not detectable by metagenomic sequencing were removed (see Figure S1C [bottom right hand corner of the graph], n=2 Cx+/EIA+ samples and n=? Cx+/EIA- samples)? The burdens of C. difficile in these individuals seem to be at least an order of magnitude lower than patients with low-but-detectable burdens and appear to be bimodally distributed. If there is no change to the conclusions of the study if these samples are removed, are there features (metabolites, taxa) that differentiate individuals who do or do not have metagenomically detectable C. difficile?

Line 144. There is no Figure S1D in the manuscript.

Lines 228-230. I really like the idea that rhamnose may be the by-product of microbiome metabolism or other dietary polysaccharides which could exclude C. difficile from the gut. What is known about rhamnose metabolism by gut microbes? Do any of the taxa that associate with Cx+/EIA+ conditions metabolize rhamnose? Do any of the taxa that associiate with Cx+/EIA- conditions metabolize rhamnopolysaccharides? This could be done by mining the metagenomes for CAZymes or by performing growth assays with isolates of relevant bacteria.

Lines 248-251: "The observed increased abundance of monosaccharides…" If this is true, shouldn't there be fewer sugars in both high diversity conditions and low diversity conditions? That is, in a high diversity microbiome, some microbes would metabolize complex polysaccahrides and others would cross feed on the waste products, whereas in a low diversity microbiome, there may be fewer degraders of polysaccharides, but maybe also fewer cross-feeders? Further expansion on this complexity or references to support the assertion in the sentence on lines 248-251 would be useful.

Lines 253-256: Given the previous reports cited in the manuscript and the range of metabolic capabilities illustrated by the clinical isolates used in this study, it would be worthwhile to assay the sugars from Figure 4 against strains of C. difficile commonly used in the C. difficile research community (at least, C. difficile 630 and difficile R0291).

Lines 262-264: This is really interesting. How many patients were excluded? Assaying toxin production in these patients (see comment on Lines 67-69, above) before and after diagnosis, or sequencing C. difficile from these patients at each of these time points, would be helpful in thinking about the transitions between asymptomatic carriage and symptomatic CDI. This is, of course, beyond the scope of the current study and only mentioned because it would be of scientific and clinical interest.

Line 320. How many Cx+/EIA- samples were included in the study?

Line 397. Please clarify in this section of the same conditions were used for VPI10643.

Lines 438-441. What percent of the variation in the data are expressed in this beta diversity metric? Is the conclusion different if a phylogenetically-aware metric (e.g. Weighted/Unweighted UniFrac) is used?

Figure S1. Please include p values for taxa noted in Figure S1C. How many patients' stool samples have undetectable C. difficile via metagenomic analysis?

Figure 4. OD readings are typically plotted on a log scale for growth curve data.

---

## [Author Response]

Essential revisions:1) Need to discuss the ability to exclude alternative hypotheses, including variations between C. difficile strains, dietary intake, differences in host physiology, and bile acid production/metabolism. The former seems like a critical point – are these individuals colonized by similar strains of C. difficile? Are they all toxin positive? It is critical to test if C. difficile from the Cx+/EIA- samples are actually capable of producing toxin. This is important to discern whether there are facets of the microbiome/metabolome which turn toxin off in Cx+/EIA- samples or if C. difficile in these patients have mutations which make them unable to produce toxin.

We thank the reviewers for their critical insight into the complexity of CDI. To address the points raised, we analyzed the ribotype distribution across *C. difficile* isolates in our Cx+/EIA- and Cx+/EIA+ patients and also used diagnostic PCR data to characterize the presence or absence of the binary toxin loci (*cdtAB*). We found a significant enrichment of the binary (CDT) toxin loci in isolates of Cx+/EIA+ patients and a significant enrichment of isolates from the RT027 lineage, considered a hypervirulent lineage of *C.*

*difficile* (Figure 1A). We also found that CDT+/EIA+ stools had slightly increased abundances of *C. difficile* (Supplementary Figure 2C, P=0.2); this is consistent with previous data that suggests that the presence of the binary toxin is associated with severe disease [1]. We incorporated these pathogen variables in revised community analyses (Figure 1, Supplementary Figure 1, Supplementary Figure 2).

Clinically, toxigenic *C. difficile* is defined as a *C. difficile* isolate with PCRdetectable *tcdAB*. Yet, as the reviewer has indicated it is very possible that the isolate has the locus present but does not express the toxin due to multiple possible levels of genetic/epigenetic variation. To answer the question of whether the Cx+/EIA- express toxin (TcdA and TcdB), we performed in vitro toxin ELISAs on all 102 *C. difficile* isolates for which we had corresponding stool metagenomic data. We found that approximately half of the isolates expressed detectable levels of toxin, and found no significant differences in the proportion of isolates that express toxin between the two study groups (Supplementary Figure 1A, P=0.86, Fisher’s exact test). As has previously been reported [2, 3], toxin expression is highly regulated by environmental cues, and can be difficult to detect during in vitro broth culture. While our in vitro data would indicate that there is no difference in the toxin expression capacity of isolates from the two cohorts, future experiments, outside the scope of this paper, will examine the variability in clinical isolate toxin expression in vivo and under different environmental cues.

2) Given the compositional nature of the sequencing data it is possible that differences in C. difficile are responsible for some of the observed differences in community structure. Please mask C. difficile reads and re-run the key analyses to check if they hold up.

We appreciate the reviewer’s statistical concern, and believe that these concerns stem from the apparent bimodality of *C. difficile* abundance. We addressed this concern both by performing analyses that the reviewer asked for, as well as by using an independent metagenomic taxonomic classifier which is more sensitive to *C. difficile* reads.

We masked *C. difficile* abundance (derived from Metaphlan metagenomic classification) and renormalized the taxonomic relative abundances to understand whether this changed the outcome of analyses from Figure 1. We found that beta diversity (using UniFrac distances) was still not significantly different between EIA+ and EIA- metagenomes (PERMANOVA, P=0.2). We performed MaAsLin2 linear mixed modeling on this dataset and found that the same commensal clostridia (from the *C. difficile* unmasked analysis) were associated with EIA- patients (Author response image 1).

**Author response image 1. sa2fig1:** Maaslin2 analysis of metagenomes with *C. difficile* reads removed and compositional dataset renormalized.

In parallel, we also used an alternate metagenomic taxonomic sequence classifier, Kraken, which uses a k-mer based approach to map reads to marker sequences. This method is more sensitive than MetaPhlAn2, and accordingly detected more *C. difficile* abundance in our NAAT+ metagenomic samples (finding *C. difficile* in 101/102 samples vs. 70/102 samples from MetaPhlAn2). Importantly, however, we observed highly correlated measures of *C. difficile* abundance (Pearson’s rho=0.94) from the MetaPhlAn2 and Kraken abundance calculations. Furthermore, we found that *C. difficile* and a number of commensal clostridia from the *Eubacterium* and *Anaerostipes* were significantly associated with EIA status in both MetaPhlAn2 and Kraken analyses (Figure 1 and Supplementary Figure 2). We were also able to demonstrate that the magnitude of correlations between *C. difficile* and metabolites was preserved between both analyses. Briefly, in our MetaPhlAn2 analysis, *C. difficile* was positively correlated with 5-aminovaleric acid and 4-methylpentanoic acid (rho=0.48 and 0.36, respectively), and negatively correlated with fructose, rhamnose, and hydroxyproline (rho=-0.27, -0.36, and -0.34, respectively). Using Kraken metagenomic taxonomic classifications (which detects more *C. difficile* in our stool microbiomes), *C. difficile* was also well-correlated with 5-aminovaleric acid and 4-methylpentanoic acid (rho of 0.51 and 0.41 respectively), and anticorrelated to fructose, rhamnose, and hydroxyproline (rho of -0.29,-0.35 and -0.37 respectively). These results are displayed in Figure 4 and Supplementary Figure 5.

Based on these findings, we do not believe that the bimodality of *C. difficile* abundances observed using MetaPhlAn2 has a major influence on our findings.

3) Please discuss the literature precedent for C. difficile growth on different carbohydrates and ideally include data for the type strain.

We have included a discussion of the literature, concerning sorbitol growth and growth on other carbohydrates to the Discussion (lines 311-313), and highlight that a number of these substrates have not been rigorously validated in clinical isolates. Additionally, we used *C. difficile 630* as the reference strain for this experiment (Figure 3B,C).

4) Please check if the conclusions are impacted by removing the Cx+/EIA- samples with metagenomically undetectable C. difficile from the computational analyses used in Figures 1-3. The concern is whether these samples are driving the perceived differences between Cx+/EIA+ patients and Cx+/EIA- patients (does C. difficile abundance or metabolite abundance still differentiate Cx+/EIA+ patients from Cx+/EIA- patients?).

We thank this reviewer for bringing up this point, and we have addressed it in detail above (Essential revision #2). Per the reviewer’s suggestion, we removed samples that had no metagenomically detectable *C. difficile* (using the MetaPhlAn2 classifier, leaving 70 samples) and reran analyses in Figure 1 to assess if the microbiome differences were preserved. We expected that this biased removal of a substantial number of samples (the samples with the lowest *C. difficile* abundance) would reduce statistical power in our analyses due to uneven reduction of sample size of our cohort, and would likely primarily affect the significance of the association with *C. difficile* abundance in our cohorts. As predicted, when we perform the MaAsLin2analyses on this reduced dataset, *C. difficile* was no longer a predictor of EIA status (FDR<0.25) by this analysis (Author response image 2). However, *Anaerostipes hadrus* and *Lachnospiraceae_bacterium_5_1_63FAA* were still the most predictive taxa associated with EIA state (with equivalent FDR values of 0.0033 and 0.0077, respectively).

**Author response image 2. sa2fig2:** Maaslin2 analysis of stool metagenomes with detectable *C. difficile.*</Author response image 2 title/legend>.

However, as we have clarified in the results, all stools were found to contain viable *tcdAB*-positive *C. difficile* isolates via selective culture, and therefore it appears the stool samples with the lowest *C. difficile* abundance are below the metagenomic sequence limit of *C. difficile* detection using MetaPhlAn2. We hypothesized that if we used an alternative taxonomic classifier with a lower limit of detection, we might be able to detect *C. difficile* in the same metagenomic data. Accordingly, we performed the same computational analyses using Kraken taxonomic classifications instead, which detected *C. difficile* in 101/102 samples vs. 70/102 samples from MetaPhlAn2. We detail the results of these analyses above and in Figure 1, Supplementary Figure 2, Supplementary Figure 5, and Figure 3, showing that our major conclusions remain unchanged from those based on Metaphlan 2 classifications. Based on our culturing data and Kraken analyses, which we explain in our revised manuscript (lines 178-179, lines 262-270), we believe it is appropriate to retain all samples in our analyses.

Reviewer #1:[…]Comments for the authors:1. Need to discuss the ability to exclude alternative hypotheses, including variations between C. difficile strains, dietary intake, differences in host physiology, and bile acid production/metabolism. The former seems like a critical point – are these individuals colonized by similar strains of C. difficile? Are they all toxin positive? I was unclear how asymptomatic carriage is defined, this is critical to the current paper and should be included in the main text and methods, not as a citation.

We appreciate the reviewer’s critical insight into a weakness of our original submission: our lack of consideration/discussion of other factors contributing to *C. difficile* pathogenesis, which we have striven to address in our revision. To clarify the basis of our comparisons, we have added statements in the Introduction and expanded on our definition of asymptomatic carriage in the beginning of the Results section, including the diagnostic pathways that led to identification of these strains as toxigenic *C. difficile* (lines 119-126). We focused on the most accessible data to us, namely data on the strain type and toxin identity of the pathogen. We performed a series of analyses on the microbiome data to assess the contribution of strain identity (using the proxy of ribotype and *cdtAB* toxin allele) to microbial community structure and present these figures in the supplementary data (Figure 1 and Supplementary Figure 1); there we highlight critical differences between strain type and *cdtAB* allele status in EIA cohorts. Additionally, we included a paragraph in the Discussion concerning the contribution of strain heterogeneity to variation in clinical outcome (lines 302-317). Finally, we discuss our limited capacity to account for dietary intake and host physiology in the Discussion section (lines 290-294).

2. Given the compositional nature of the sequencing data it is possible that differences in C. difficile are responsible for some of the observed differences in community structure. I'd recommend masking C. difficile reads and re-running the key analyses to check if they hold up.

We addressed this response in detail above (Essential revision 2). Briefly, we masked the reads, renormalized the relative abundances for each patient, and revealed that commensal Clostridia species were still significantly associated with Cx+/EIA- patients.

3. The in vitro validation is helpful, but I'm unclear as to whether it is new information. If any prior studies have been done they should be cited here.

We thank the reviewer for this consideration and acknowledge that while information on fructose and sorbitol usage have been studied in reference or commonly used strains, here we present new data on nutrient utilization by clinical isolates. This revealed the while some strains appear to grow as robustly as the reference isolates (*C. difficile* VPI10643 and *C. difficile* 630), some do not appear to grow as well on sorbitol as the sole carbon source. Fructose and sorbitol served as well-known utilizable carbohydrate substrates such that we could draw conclusions about the ability of *C. difficile* to utilize carbohydrates such as lactulose, rhamnose, and sucrose. Finally, we have added to the Discussion section (lines 311-314) to highlight the preceding literature to our work and the limitations of our nutrient utilization data.

Reviewer #2:[…] Comments for the authors:Given that this manuscript is primarily computational, it would be beneficial if the code for the analysis was shared in a public repository.

We have generated a.Rmd file and deposited this to Github.

PRJNA748262 does not appear to be publicly available.

We have made this data public.

Line 66: would asymptomatic colonization be on the disease spectrum?

We have revised this wording to clarify this point, specifically by changing the phrase “disease spectrum” to “clinical manifestation” (line 72).

Reviewer #3:[…] Lines 67-69. What is the limit of detection of EIA? It's striking to me that, while C. difficile burdens are a predictor of Cx+/EIA+ vs. Cx+/EIA- patients, there are ~23 Cx+/EIA- patients whose relative abundance of C. difficile essentially superimposes with those of the Cx+/EIA+ group. There are fibroblast rounding assays that quantify C. difficile toxins. The limit of detection of these assays may be lower than kits used in diagnostic labs. Can the authors perform one of these assays on the samples in this study to better calibrate readers expectations of whether toxigenic C. difficile is or is not expressing toxin? As the paper stands, I assume that production of toxin in C. difficile is all-or-nothing, but prior work on regulation of toxin production in C. difficile has demonstrated that many regulatory cues influence toxin production (Martin-Verstraete et al. 2016) so I expect this to be non-binary. In addition, and considering the regulatory networks that control toxin production, are the C. difficile isolates from asymptomatic patients capable of producing toxin in vitro? Just because the gene is there doesn't mean that it's functional.

We sincerely appreciate this concern and have addressed it in detail (Essential revision #1). Briefly, our in vitro ELISA data indicates that half of isolates were positive for the toxin (this frequency was not different between EIA- and EIA+, Supplementary Figure 1A). This data suggests that there is no difference in the number of isolates that produce toxin, and namely, that a significant proportion of EIA- isolates are able to express toxin in vitro*.* However, as the reviewer is no doubt aware, this is an inadequate method of assessing a strain’s in vivo capacity to elaborate toxin [2, 3]. We have highlighted this limitation in our discussion (lines 310-313).

Line 140-142. "Given that C. difficile levels were an overt predictor of CDI…" Are C. difficile levels still a predictor of the Cx+/EIA+ state if the samples where C. difficile was not detectable by metagenomic sequencing were removed (see Figure S1C [bottom right hand corner of the graph], n=2 Cx+/EIA+ samples and n=? Cx+/EIA- samples)? The burdens of C. difficile in these individuals seem to be at least an order of magnitude lower than patients with low-but-detectable burdens and appear to be bimodally distributed. If there is no change to the conclusions of the study if these samples are removed, are there features (metabolites, taxa) that differentiate individuals who do or do not have metagenomically detectable C. difficile?

We have addressed this in Essential revision 2 and 4. If samples with metagenomically undetectable levels of *C. difficile* are removed, as expected, the *C. difficile* association was no longer statistically significant. Based on our culture data and the use of an alternative metagenomic taxonomic classifier (Kraken), we hypothesize that samples with no metagenomically detectable *C. difficile* (by MetaPhlAn2) are those that contain the lowest abundance of the organism. To support the conclusions of our manuscript, we performed identical analysis of differentially abundant taxa (using Metaphlan in Figure 1 and Kraken in Figure S2), and found that increased *C. difficile* is associated with Cx+/EIA+ patients and a number of commensal Clostridia are associated with Cx+/EIA- patients in both analyses of the same sequencing data. Additionally, we performed identical analyses of multi-omic correlates of these cohorts (Figure 3 and Figure S5) and found the findings were preserved. Briefly, in our Metaphlan analysis, *C. difficile* was positively correlated with 5-amino-valeric acid and 4-methylpentanoic acid (rho=0.48 and 0.36, respectively), and negatively correlated with fructose, rhamnose, and hydroxyproline (rho=-0.27, -0.36, and -0.34, respectively). Using Kraken-classified metagenomic data (which detects more *C. difficile* in our stool microbiomes), *C. difficile* was well-correlated with 5-amino-valeric acid and 4-methylpentanoic acid (rho of 0.51 and 0.41 respectively), and anti-correlated to fructose, rhamnose, and hydroxyproline (rho of -0.29,-0.35 and -0.37 respectively).

Line 144. There is no Figure S1D in the manuscript.

We have corrected this error.

Lines 228-230. I really like the idea that rhamnose may be the by-product of microbiome metabolism or other dietary polysaccharides which could exclude C. difficile from the gut. What is known about rhamnose metabolism by gut microbes? Do any of the taxa that associate with Cx+/EIA+ conditions metabolize rhamnose? Do any of the taxa that associate with Cx+/EIA- conditions metabolize rhamnopolysaccharides? This could be done by mining the metagenomes for CAZymes or by performing growth assays with isolates of relevant bacteria.

In response to multiple reviewer and editorial comments, we have stepped back from hypotheses concerning specific metabolites. Given the complexity of CDI and the cross-sectionality of our data, we have refocused the manuscript more broadly on trying to understand asymptomatic colonization using pathogen and microbiome data. Yet, we felt it was an important exercise to examine the presence of CAZymes in our metagenomic data. Thus, we utilized Humann2 functionally-annotated gene family abundance data, and identified all KEGG-annotated genes with the enzyme commission number EC 3.2.1 [4], as these represent glycoside hydrolases that could be involved in carbohydrate degradation. We found a number of genes putatively involved in starch and sucrose metabolism that were increased in EIA- patients, supporting our MetaCyc pathway analysis in Figure 2. We are extremely interested in identifying species metabolite utilization profiles, specifically in the case of starch and rhamnopolysaccharides, but feel that it is outside the scope of this manuscript. Yet, we have added a speculative note concerning Clostridia metabolism to the Discussion section (line 348-349).

Lines 248-251: "The observed increased abundance of monosaccharides…" If this is true, shouldn't there be fewer sugars in both high diversity conditions and low diversity conditions? That is, in a high diversity microbiome, some microbes would metabolize complex polysaccahrides and others would cross feed on the waste products, whereas in a low diversity microbiome, there may be fewer degraders of polysaccharides, but maybe also fewer cross-feeders? Further expansion on this complexity or references to support the assertion in the sentence on lines 248-251 would be useful.

We are extremely thankful for the reviewer’s insight and curiosity concerning the relationship between monosaccharide levels and microbiome diversity. Based on these comments, we hypothesized that there might be a relationship between monosaccharide levels and microbiome diversity. To directly examine this relationship, we summed levels of representative (and biochemically confirmed) monosaccharides: fructose, rhamnose, and glucose. We then measured correlation between the sugar sum and 3 different measures of alpha diversity and found no strong correlation, as measured by Pearson’s rho: Shannon diversity(-0.083), richness(-0.045), and Faith’s diversity (-0.081). We have also stepped back from speculating as to the meaning behind the level of monosaccharides. As we have not accounted for differences in host diet, we note that we cannot attribute the differences in levels monosaccharides to differences in host diet or differences in cross-feeding rate in the community (lines 299-301).

Lines 253-256: Given the previous reports cited in the manuscript and the range of metabolic capabilities illustrated by the clinical isolates used in this study, it would be worthwhile to assay the sugars from Figure 4 against strains of C. difficile commonly used in the C. difficile research community (at least, C. difficile 630 and difficile R0291).

We have redone these experiments with *C. difficile* 630, a commonly used reference strain, and also kept *C. difficile* VPI10643, which is a commonly used reference strain for animal experiments [5, 6]

Lines 262-264: This is really interesting. How many patients were excluded? Assaying toxin production in these patients (see comment on Lines 67-69, above) before and after diagnosis, or sequencing C. difficile from these patients at each of these time points, would be helpful in thinking about the transitions between asymptomatic carriage and symptomatic CDI. This is, of course, beyond the scope of the current study and only mentioned because it would be of scientific and clinical interest.

We appreciate the reviewer’s excitement, and we are in the process of trying to study such transitions both in animal models and in patients.

Line 320. How many Cx+/EIA- samples were included in the study?

We have added numbers to the text for precision.

Line 397. Please clarify in this section of the same conditions were used for VPI10643.

We have clarified this in the text.

Lines 438-441. What percent of the variation in the data are expressed in this beta diversity metric? Is the conclusion different if a phylogenetically-aware metric (e.g. Weighted/Unweighted UniFrac) is used?

More variation is incorporated in the principal coordinate analysis of weighted UniFrac distance (a phylogenetically-aware metric, Author response image 3), compared to that of Bray Curtis dissimilarity (Author response image 3). We had previously quantified beta-dispersion between groups, but found that it would be more appropriate to quantify the permutational analysis of variance (PERMANOVA) using the adonis package and found that using phylogenetically-aware measures such as unweighted UniFrac distance and weighted UniFrac dissimilarity yielded insignificant differences in community structure (P=654, P=0.233), compared to that of Bray Curtis (P=0.415). For this reason, we have decided to present weighted Unifrac dissimilarity in the main text.

**Author response image 3. sa2fig3:** 

Figure S1. Please include p values for taxa noted in Figure S1C. How many patients' stool samples have undetectable C. difficile via metagenomic analysis?

We have included p-values. Additionally, 32 out of 102 samples have undetectable C. difficile via MetaPhlAn2 metagenomic analysis; however, please see response to Essential revision #2 above details regarding culture-based *C. difficile* detection and Kraken-based metagenomic detection (*C. difficile* detected metagenomically in 99% of samples).

Figure 4. OD readings are typically plotted on a log scale for growth curve data.

We have redone this figure and plotted it on a log-scale.

References:

1. Cowardin, C.A., E.L. Buonomo, M.M. Saleh, M.G. Wilson, S.L. Burgess, S.A. Kuehne, C. Schwan, A.M. Eichhoff, F. Koch-Nolte, D. Lyras, K. Aktories, N.P. Minton, and W.A. Petri, Jr., The binary toxin CDT enhances Clostridium difficile virulence by suppressing protective colonic eosinophilia. Nat Microbiol, 2016. 1(8): p. 16108.

2. Burnham, C.A. and K.C. Carroll, Diagnosis of Clostridium difficile infection: an ongoing conundrum for clinicians and for clinical laboratories. Clin Microbiol Rev, 2013. 26(3): p. 604-30.

3. Akerlund, T., B. Svenungsson, A. Lagergren, and L.G. Burman, Correlation of disease severity with fecal toxin levels in patients with Clostridium difficile-associated diarrhea and distribution of PCR ribotypes and toxin yields in vitro of corresponding isolates. J Clin Microbiol, 2006. 44(2): p. 353-8.

4. Tanes, C., K. Bittinger, Y. Gao, E.S. Friedman, L. Nessel, U.R. Paladhi, L. Chau, E. Panfen, M.A. Fischbach, J. Braun, R.J. Xavier, C.B. Clish, H. Li, F.D. Bushman, J.D. Lewis, and G.D. Wu, Role of dietary fiber in the recovery of the human gut microbiome and its metabolome. Cell Host Microbe, 2021. 29(3): p. 394-407 e5.

5. Theriot, C.M., C.C. Koumpouras, P.E. Carlson, Bergin, II, D.M. Aronoff, and V.B. Young, Cefoperazone-treated mice as an experimental platform to assess differential virulence of Clostridium difficile strains. Gut Microbes, 2011. 2(6): p. 326-34.

6. Erikstrup, L.T., M. Aarup, R. Hagemann-Madsen, F. Dagnaes-Hansen, B. Kristensen, K.E. Olsen, and K. Fuursted, Treatment of Clostridium difficile infection in mice with vancomycin alone is as effective as treatment with vancomycin and metronidazole in combination. BMJ Open Gastroenterol, 2015. 2(1): p. e000038.